## Evidence synthesis

behaviour, bioengineering, health and disease and epidemiology

disease, distress, precision livestock farming, animal behaviour, early detection

**Author for correspondence:**
Joanna Stachowicz
e-mail: joanna.stachowicz@agroscope.admin.ch

# Do we automatically detect health- or general welfare-related issues? A framework

Joanna Stachowicz and Christina Umstätter

Research Division on Competitiveness and System Evaluation, Agroscope, Tänikon 1, 8356 Ettenhausen, Switzerland

 JS, 0000-0001-5373-3855; CU, 0000-0002-7638-3867

The early detection of health disorders is a central goal in livestock production. Thus, a great demand for technologies enabling the automated detection of such issues exists. However, despite decades of research, precision livestock farming (PLF) technologies with sufficient accuracy and ready for implementation on commercial farms are rare. A central factor impeding technological development is likely the use of non-specific indicators for various issues. On commercial farms, where animals are exposed to changing environmental conditions, where they undergo different internal states and, most importantly, where they can be challenged by more than one issue at a time, such an approach leads inevitably to errors. To improve the accuracy of PLF technologies, the presented framework proposes a categorization of the aim of detection of issues related to general welfare, disease and distress and defined disease. Each decision level provides a different degree of information and therefore requires indicators varying in specificity. Based on these considerations, it becomes apparent that while most technologies aim to detect a defined health issue, they facilitate only the identification of issues related to general welfare. To achieve detection of specific issues, new indicators such as rhythmicity patterns of behaviour or physiological processes should be examined.

## 1. Introduction

The early detection of health disorders is one of the paramount aims in today's livestock industry, as it is a key factor for reducing the suffering of animals. On-farm assessment is generally accomplished by observing the animals directly by farm staff. However, direct observations are time-consuming, subjective and at risk of delayed detection. Another obstacle might be posed by the presence of humans during the inspection as the probability of animals trying to mask any vulnerability also increases [1]. The use of automated technologies enabling the real-time monitoring of animals can provide support in this matter.

Such technologies are subsumed under the term precision livestock farming (PLF) and are composed of hardware and intelligent software, which collect and analyse data of animals and/or their environment [2]. However, despite years of research and numerous studies focusing on the development or validation of PLF technologies for the assessment of health-related changes, to date only a few systems are commercially available [3]. Further, the implementation rate of the available systems on commercial farms is in some countries strikingly low [4,5]. The discrepancy between commercialization, implementation and research can be largely attributed to the missing breakthrough in reliable predictive models and a sound return on investment.

This statement seems to contradict the seemingly sufficient performance of PLF technologies presented in many studies [6]. The upper values obtained for specificity (model's ability to detect negative outcomes) found in the literature range around

88.8% for lameness [7], 97.4% for respiratory diseases [8], 87% post-calving diseases [9] and 84.1% for mastitis [10]; for sensitivity (model's ability to detect positive outcome), they range around 82.3% for mastitis [10], 100% for respiratory diseases [8], 69% post-calving diseases [9] and for lameness 86.1% [11]. However, high values of specificity and sensitivity can be achieved despite high error rates [6] and a considerable number of false-positive alerts [12]. On commercial farms, a high number of false-positive alarms is especially disadvantageous as it disrupts the daily management routine and ultimately leads to the non-compliance with alarms by the farmer [6]. As a result, the performance of PLF technologies cannot be judged by the values of sensitivity and specificity alone as these measures do not properly reflect the suitability of PLF technologies for implementation on commercial farms [6,12].

When aiming to develop PLF technologies for the assessment of disease-related changes in livestock for commercial farms, one faces numerous challenges related to data collection, data analysis and technological aspects. An important consideration for data collection is the study design, which usually implies a trade-off between controlled conditions and a low external validity and uncontrolled conditions characterized by a high variability and, in return, a high external validity. In addition, the identification of suitable indicators for the respective purpose is another crucial factor. For data analysis, a variety of approaches (e.g. multivariate cumulative sum control charts [13], wavelet filtering analysis [14], time series analysis [15] or fuzzy logic [10]) and performance measures (e.g. specificity, sensitivity, success rate, receiver operating curve or area under curve [6]) are available, and the most appropriate ones need to be determined. Further, in the scope of development, one also encounters technological difficulties. For example, PLF technologies have to function in a harsh and unfavourable farming environment, coping with dust, high ammonium concentrations, humidity or rough physical impact from animals.

To accelerate development and increase the adoption rate of suitable PLF technologies, new approaches are needed to overcome all the mentioned challenges. In this paper, we present a framework for the development of PLF technologies for the early detection of issues in livestock by focusing on the decision process related to data collection. On the basis of recent studies, underlying questions regarding the aim of detection and the suitability of variables serving as indicators in view of a commercial farm setting will be critically reviewed. Then a decision pathway will be proposed, which detangles the aim of detection and guides the user through the different decision levels and their options by pointing out the benefits and disadvantages of each.

## 2. Methods

### (a) Literature research for identifying challenges in studies on precision livestock farming technologies for the automated detection of health-related issues in livestock

To identify the obstacles and challenges in the development and validation of PLF technologies for the automated assessment of health-related issues in livestock, we reviewed the methodological approaches linked to data collection, which have been used so far.

**Table 1.** Search terms used in the present review to identify literature on the automated assessment of health-related issues in livestock.

| enterprise terms | precision livestock farming terms | type of issue terms |
| --- | --- | --- |
| dairy cow | smart sensor | disease |
| cattle | smart farming | health disorder |
| calves | automated monitoring | stress |
| pig | precision livestock farming | behaviour |
| sow | | |
| broiler | | |
| laying hen | | |
| goat | | |
| sheep | | |

To do so, we have conducted a systematic literature search based on the PRISMA guidelines [16]. Two databases have been used, PubMed and Scopus. The search fields were 'article title, abstract, keywords' in Scopus, and 'all fields' in PubMed. In addition, to exclude human studies, the category 'other animals' was chosen in PubMed. Fourteen search terms were used in total (table 1). Each search string was composed of an 'enterprise' term, a 'precision livestock farming' term and a term for the 'type of issue'. Between each of the terms, the Boolean operator 'AND' was set; for example, 'dairy cow' AND 'automatic monitoring' AND 'disease'. The combinations of the different terms of the three groups resulted in 144 search strings. The search protocols can be found in the electronic supplementary material, table S1.

In addition, Google Scholar was used to find studies that were identified based on the reference hits but could not be found in the other two databases. Most full texts have been retrieved via Science Direct. The related article suggestions provided by Science Direct resulted in the identification of further literature.

To identify relevant studies within the hits, an eligibility screening was conducted using the following inclusion and exclusion criteria.

Inclusion criteria:

— Studies focusing on the development or validation of technologies or models for the automatic assessment of the animal's state or an issue, including oestrus, parturition, defined and general diseases, distress and abnormal behaviours for the enterprises of dairy cows, cattle, calves, pigs, sows, broilers, laying hens, goats and sheep.
— Studies with positive and negative results.
— Studies written in English.
— Studies published at any time.

Exclusion criteria:

— Studies focusing solely on the technical performance of PLF technologies or in other words studies that tested the feasibility to record the aimed variables, without aiming to link the recorded variables to the animal's state or issue.
— Studies looking for a relationship between stressors and the animals' responses, but again without the intention to use the variables for issue detection.
— Reviews.

The literature screening was done based on the titles and abstracts. To reduce the risk of inconsistent interpretation of the inclusion and exclusion criteria [17], one researcher performed the screening.

## (b) Developing a guide for the automated assessment of issues in livestock

After the identification of the obstacles and challenges in the development and validation of PLF technologies for the automated detection of health-related issues, in the next step, the aim was to detangle the aim of detection based on the degree of information. Three decision levels of issue detection were proposed. For each decision level, a separate literature review was conducted by screening the previously mentioned databases. Because the three proposed decision levels cover a range of disciplines, as a final but essential step, a focus group consisting of six experts specialized in the fields of agricultural engineering, animal behaviour, applied animal behaviour and veterinary science was held with the purpose of critically discussing and verifying the proposed framework. At first, each of the experts reviewed the manuscript on its own and then all comments and suggestions were discussed with all experts in a meeting. Based on the discussion, three major adjustments were decided for the manuscript. (i) To provide a wider overview of the different challenges in the development of PLF technologies in the introduction, including technical difficulties and methods for data analysis. (ii) To improve the terminology of veterinary terms in the 'Defined disease-related issues' section. (iii) To add a discussion to the synthesis about the current expectation towards PLF technologies and what actually can be accomplished to date, under consideration of the criteria proposed in the framework.

## 3. Results

For qualitative synthesis, 100 articles were found. The screening and selection process is shown in figure S1, in electronic supplementary material. The identified literature represents a sample of evidence composed of peer-reviewed studies ($n = 80$) and conference proceedings ($n = 20$). In 64 cases, the aim of detection was disease related, whereas 36 studies focused on the detection of thermal and behavioural issues. More than half of the studies ($n = 57$) used a non-specific approach for the detection of a specific issue, 12 studies used an intermediate and only 18 followed a more specific approach. The distribution of studies based on the year of publication is presented in table 2. The full list of studies is provided in electronic supplementary material, table S2.

## 4. Framework

## (a) Identifying challenges in the development of precision livestock farming technologies for assessing issues in farm animals

PLF technologies can support animal health provided the systems reliably identify any related changes [6]. Thus, over the last decades, a growing number of studies have focused on the development or validation of PLF technologies for the detection of health-related issues in livestock e.g. [18–21]. Some studies intended to identify a general health disorder [22–24]; some concentrated on defined health disorders, such as mastitis [25] or ketosis [26], while others focused on clinical signs of disease, such as lameness [27] or coughs [28]. However, despite the different aims, in many cases, the same variables were arbitrarily applied, which is illustrated in the following examples. Timsit *et al.* [29] used reticulorumen temperature as an indicator for respiratory disease in young

**Table 2.** Number of publications by year, includes publications up to the end of April 2020.

| year published | number of publications |
| --- | --- |
| 1999 | 1 |
| 2005 | 1 |
| 2007 | 4 |
| 2008 | 2 |
| 2009 | 4 |
| 2010 | 1 |
| 2011 | 2 |
| 2012 | 3 |
| 2013 | 8 |
| 2014 | 2 |
| 2015 | 6 |
| 2016 | 10 |
| 2017 | 21 |
| 2018 | 12 |
| 2019 | 19 |
| 2020 | 4 |

bulls, while Adams *et al.* [19] used it for the detection of mastitis, pneumonia, metritis and lameness in dairy cows. Changes in milk yield, rumination and neck activity were shown to serve as early signs of lameness [30], whereas, rumination and activity served also for the identification of metritis [31], metabolic digestive disorders [32] and mastitis [25]. Moreover, changes in activity also indicated the occurrence of lameness in broilers [18] and in sheep [33]. Lowe *et al.* [34] found that milk consumption, body temperatures of the side and shoulder and number and duration of lying bouts have the potential to be suitable measures of neonatal calf diarrhoea. And Steensels *et al.* [9] used rumination, activity, milk yield, body weight and voluntary visits to the milking robot to detect post-calving diseases. All the presented variables are non-specific and can change under different conditions and with various physiological states, health- and behaviour-related issues. This becomes clear when looking at the following studies. Abeni al *et al.* [35] used activity and rumination time to identify heat stress in dairy cows, while Rutten *et al.* [36] used activity, rumination and ear temperature to predict the start of calving. Further, Wallenbeck & Kneeling [37] used the frequency of visits to the electronic feeders and feed consumption to indicate tail-biting outbreaks in pigs. Finally, sensors recording activity, rumination and reticulorumen temperature were also applied to detect changes around oestrus [38],

Thus, using non-specific indicators for a specific issue such as a defined disease or a symptom should have a negative impact on the accuracy of the predictive value. Although many of these studies could prove the value of the variables used as reliable indicators for the detection of the issue in focus, it stands to reason that the results might be misleading as it is unlikely that such relationships can always be established under on-farm conditions. This is because in a commercial farm setting animals can experience more than one state at a time, and using non-specific variables could

lead to a high number of false-positive results. In fact, Dominiak & Kristensen [6] stated that for two decades, no predictive model has reached the performance demands needed to generate a satisfactorily low number of false-positive alarms. As the development of suitable PLF technologies begins with appropriate measures [39], it is essential to consider exactly what particular issue is to be detected and which variables are suitable as indicators for the respective purposes. The more specific the aim is, the more precise the variables or combination of variables need to be to improve rates of successful detection. Although this conclusion seems to be obvious and therefore redundant, the minority of studies followed this approach. For example, a few of the exceptions are the studies by Maselyne et al. [40], who used feeding-related variables for detecting any kind of issue (health-, welfare- or production-related) in pigs, by Silva et al. [20] who investigated cough sounds of infected and non-infected pigs and by Villot et al. [41] who used reticulorumen pH as an indicator for subclinical rumen acidosis. There are also studies, which used an intermediate approach in regard to the specificity of indicators. For example, Viazzi et al. [42] tried to identify lameness in cows based on the back posture. While an arched back can be also a sign of general pain [43] or for other health disorders such as traumatic reticuloperitonitis and abomasal ulcers [44], it is at least a good indicator for health-related issues and unlikely caused by other stressors such as social or thermal stress.

One reason for the use of variables that are not exclusively related to the issue in focus is that the application of specific variables is limited by technological development. In fact, besides a few systems, such as SoundTalks (SoundTalks NV, Leuven, Belgium), which monitors respiratory issues by recording cough sounds, or rumen boluses (pH Plus Bolus, smaXtec, smaXtec, Graz, Austria), which can potentially identify rumen acidosis based on rumen pH, most other animal-based sensors are able to record only basic variables, such as feeding, activity, body weight or the temperature of the animals [3]. However, in such cases where only non-specific variables are available, the aim of detection should be restricted to whether or not an issue exists without specifying the issue as this increases the error rate.

Consequently, in the scope of development, aspects such as the exact aim of detection and the specificity of indicators should be considered. Only then, one will be aware of the restrictions of the model or system and of the results which can be truly expected. In addition, it might be possible to minimize errors, reduce false-positive alerts and increase the accuracy of the predictive models.

To improve the performance of PLF detection, first, it is of utmost importance to precisely define the aim of the system to be developed. We propose a categorization of the aim into the following three levels of detection based on their degree of information: general welfare-related issues represent the first level of detection as it embodies all kinds of health- and distress-related issues (figures 1 and 2). The aim of the second level is to differentiate between health- and distress-related issues (figures 1 and 3). And finally, the third level of detection aims to identify disease-related issues and, thus, presents the most specific aim (figures 1 and 4). For each decision level, different considerations have to be factored in, leading to different decision pathways. Hence, in the next section, we will reflect on the particular challenges of each decision level and discuss the sub-levels of their respective pathways.

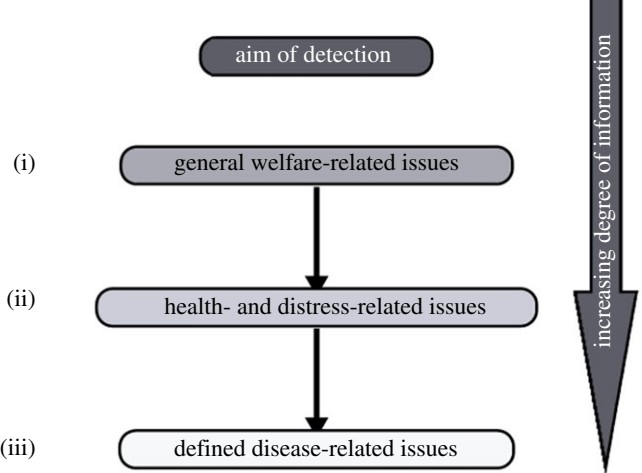

**Figure 1.** Overview of the three decision levels (i–iii) based on the degree of information.

A scheme of the proposed decision pathway with its different levels is depicted in figure 1. It has to be noted that the sub-levels of the three decision levels are interchangeable and do not follow a strict order (figures 2–4).

## (b) A guide to the automated assessment of issues in livestock at different levels of specificity using precision livestock farming technologies

### (i) General welfare-related issues

The first decision level deals with the detection of 'general welfare-related issues' (figures 1 and 2). Animal welfare is a multidimensional concept, implying that an animal is in good health, in a positive affective state and able to perform natural behaviours within its repertoire [45], which are pleasurable and promote biological function [46]. Here, the use of non-specific indicators is justifiable. That is because a non-specific indicator changes due to varying conditions and various issues and thus cannot be used on its own to detect disease-related changes, but it may be suitable to detect any disturbance may it be health- or distress-related. However, due to the non-specificity, the interpretation is only limited to whether there is a potential issue or not. Which variables are available and could potentially be used in the future will be discussed in the next sections.

### Indicators for the assessment of welfare-related issues

For the automated detection of welfare-related issues in livestock, two sets of indicators can be used, namely environment- and animal-based indicators. Environment-based can be collected using sensors, which are usually placed in the animal's environment. Their advantage is that the interpretation is straightforward because the sensors directly provide the variable(s) of interest [47]. However, they do not reflect the animal's welfare status and thus can only be seen as risk factors for welfare impairment. Nevertheless, because environmental indicators are known to influence animal-based indicators greatly [48–50], they can complement the animal-based indicators [51].

Animal-based indicators can be recorded with animal-borne sensors, such as accelerometers or RFID tags or with strategically placed systems, such as weighing systems,

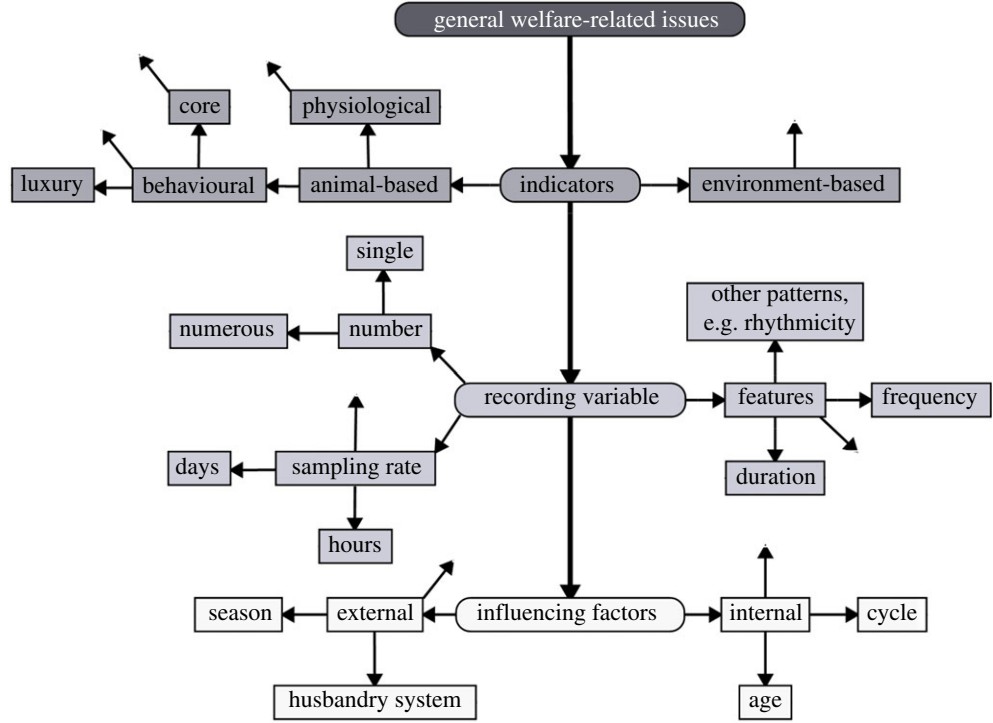

**Figure 2.** Decision pathway for general welfare-related issues (I). The free arrows indicate that there are more options than presented.

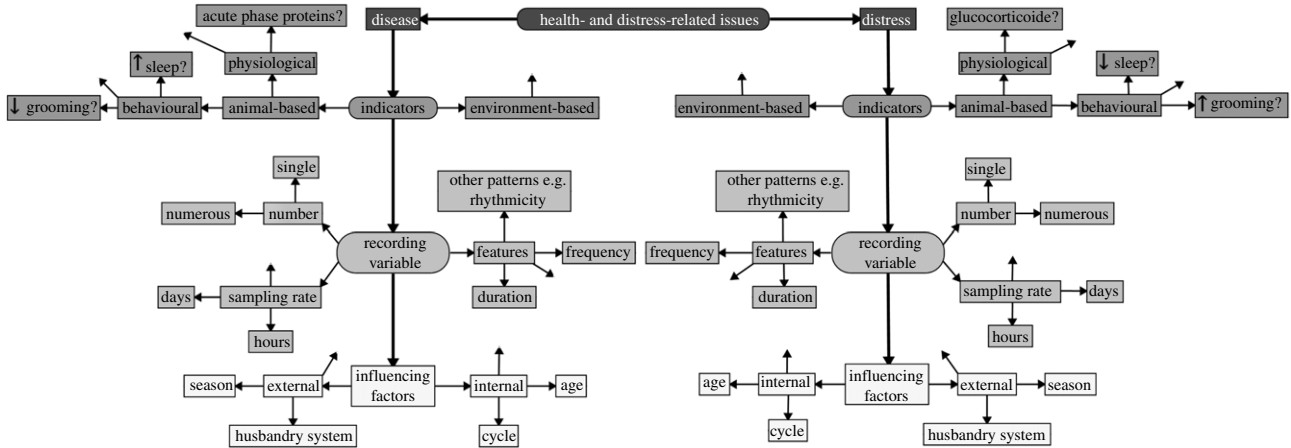

**Figure 3.** Decision pathway for health- and distress-related issues (II). The free arrows indicate that there are more options than presented.

cameras or sensors integrated into the milking parlour. However, outcome interpretation can be difficult due to the complexity of physiology and behaviour, variation over the course of time [52], the considerable intra- and inter-variability [11,53] and bidirectional changes [54–57]. Despite the potential of ambiguity, animal-based indicators are generally better measures for the detection of welfare-related issues than environment-based ones as they represent the direct responses of the animal [51].

Animal-based indicators can be further divided into behavioural and physiological indicators. The main advantage of physiological indicators is that they respond very fast to change, especially on the molecular level. This is because a reaction to any kind of stressor, be it physical or psychological, usually starts with a molecular cascade [58,59]. Unfortunately, the real-time collection and analysis of physiological indicators on the molecular level (e.g. hormones) are

not yet feasible. Despite ongoing research on biosensors for the identification of definite biomarkers, such as pathogens [60–62], cytokines [63] and glucocorticoids (GCs) [64,65], the adoption of point-of-care diagnostics in livestock, which allows the rapid detection of analytes in animals, is still a concept of the future. There are other physiological indicators though, such as body or rumen temperature, which nowadays can be measured through sensors integrated into ear tags or rumen boluses [3].

In terms of behavioural indicators, feeding, rumination and activity are most frequently used. As these behaviours serve an immediate function and thus provide short-term value [1], they are considered core behaviours [66]. The advantage of using core behaviours as indicators is that animals can express them under most circumstances, regardless of the husbandry system or environmental or group effects; only the extent and pattern of occurrence might vary between

**Figure 4.** Decision pathway for defined disease-related issues (III). The free arrows indicate that there are more options than presented.

conditions. The disadvantage is that a change in core behaviours is likely to occur later than e.g. grooming, exploration and play [1,67], which are often referred to as luxury behaviours. Luxury behaviours are considered to provide long-term benefits and thus are neglected when an animal is challenged by a stressor [66,68–70]. However, to date, they have received less attention as potential indicators. In fact, to our knowledge, there is not one commercially available PLF system, which uses luxury behaviours as indicators for the detection of welfare-related changes [3].

### Recording of indicators

Another important consideration is whether to use a single variable or multiple variables as indicators. For example, in instances where variables show an inter-correlation, such as feeding and rumination [71], or where one variable is indicative of an issue, using several might be redundant. However, there is accumulating evidence that multiple variables are likely to provide a clearer picture as to whether an issue exists [72,73]. For the application of a variety of sensors, it is a prerequisite that the sensors are affordable and that management systems are available that are capable of integrating, analysing and interpreting the data from the various sensors.

Sensors for the assessment of behavioural changes usually record different features of a given parameter, such as the frequency, duration or the occurrence during the time of day. The simultaneous recording of different characteristics has proven to be advantageous in cases where, for example, a change in a variable occurs in one feature but not in another. For instance, in cattle, a change was visible in the bout frequency and duration of lying and standing after a gastrointestinal parasite infestation but not in total duration [74]. In addition, over the years, the complexity of behaviours has been recognized, and it was postulated that detecting temporal or directional patterns [75] and sequences [76] might be more meaningful than just quantifying the frequency and duration of a given

behaviour. For example, in mice, specific characteristics such as when, how long and in which direction grooming was performed could be assigned to comfort or to distress [75]. Further, Veissier et al. [77] found that, based on circadian variation, it was possible to characterize the physiological and pathological states of a cow, such as oestrus, lameness and mastitis. The authors even concluded that circadian variation appears to be an earlier sign of an issue than changes in activity or feeding behaviour. Yet, despite the potential, particularly for the differentiation between positive and negative states, specific patterns of behaviour and rhythmicity remain largely overlooked in research to date.

To sufficiently detect changes in variables, it is also necessary to define a sampling rate. The appropriate interval depends on whether a slow (e.g. weight) or a fast (e.g. feeding, activity) changing parameter is going to be recorded. Weighing systems often record the weight of an individual once per day, whereas the sampling interval of acceleration (e.g. 4 Hz ICECUBE or 16 Hz ICETAG, IceRobotics Ltd, Edinburgh, UK) or temperature sensors (e.g. 0.001 Hz Calf Tag, FeverTags, Amarillo, USA) is much higher.

### External and internal influencing factors

Animals continuously adjust their physical and behavioural processes in response to changing external but also internal conditions [52]. For example, female cows are known to alter their behaviour when in oestrus [78], during different lactation states [40] and before parturition [71]. Depending on age, animals also vary in their general activity [79] and in their body temperature [80]. In addition, environmental factors, such as seasonal effects have an impact on the behaviour and physiology of animals. As reported by Yalcin et al. [81], the feed consumption of broilers was reduced by 23% in summer conditions when compared with autumn conditions. Further, an increase in body temperature of cows before the onset of heat was detectable in winter but not in

summer [82]. Depending on the husbandry system, animals may also vary in the strength of their responses, allowing the identification of physiological states or issues more easily in one than in another. Free-ranging hens, for example, exhibited a higher variety of sickness behaviour than hens raised in cages [83]. Since all the external and internal factors mentioned are associated with physiological changes and the expression of behaviour, it is important to take these into account when selecting indicators.

In summary, the detection of welfare-related issues requires a variety of considerations. The main advantage at this level of issue detection is that animal welfare is a multi-factorial concept and therefore a wide range of variables can act as indicators. However, the appropriate indicators need to be assessed in terms of their ambivalent meanings, their suitable combinations and their characteristics, which are going to be recorded. By considering all the presented aspects, the reliability of the detection of welfare-related changes can be enhanced. Most importantly, it can be concluded that at this decision-making level, the focus is on whether or not there is a welfare-related issue.

### (ii) Health- and distress-related issues

In the previous level of detection (figures 1 and 2), a variety of welfare-related issues were included and thus a large number of non-specific variables could be used. The aim of the second decision level is to retrieve a higher degree of information about the cause of an issue and the required interventions by differentiating between different stressors (figure 3). While the same decision pathway can be followed as in section '(i) General welfare-related issues', appropriate indicators need to be chosen, which allow discriminating between different states. To do so, we will look at the main adverse states an animal can experience in intensive housing and what potential markers might allow us to differentiate between them.

Animals in husbandry systems face a range of physical and psychological challenges. On a regular basis, they have to undergo different management procedures usually involving some kind of handling, which can elicit fear [84]. Further, animals often suffer from health disorders [85,86], can be exposed to unfavourable thermal conditions [87] and can frequently be subjected to aggressive encounters with conspecifics [88,89]. All of these challenges act as stressors as they pose a threat to the animal's homoeostasis, which is defined as the balance of bodily states [90].

Since some stress responses can mask or influence each other, distinguishing between different stressors should be beneficial. For example, chronic psychological challenges are known to increase susceptibility to infections but also negatively affect their progression and the recovery process [91,92]. Thus, by identifying issues elicited by distress, it might be even possible to prevent the development of some diseases. In addition, although considered beneficial, intense short-term psychological challenges can mask signs of diseases [93] or lead to false-positive health alarms when ignored. In turn, signs accompanying diseases such as pain or malaise might themselves be stressful for the animals. Finally, issues arising from different stressors may require different interventions. While adjusting management could be necessary for all stressors, health disorders additionally require medical treatment. Thus, in the present framework, we will refer to

responses caused by negative experiences that are emotionally (social separation: [94]; handling: [84]) or thermally [87] challenging as distress. Physiological challenges, which directly affect animal health, such as infections, traumata or inflammations, are referred to as diseases. Despite the intertwined nature of distress and disease, surprisingly, there are only a few studies focusing on the detection of distress-related changes, and there are hardly any studies that have tried to distinguish between health- and distress-related issues. In the following sections, we will look at some potential indicators, which may help to differentiate between distress and disease.

### Indicators for health and distress-related issues

To be able to distinguish between health and distress-related issues, it is important to define adequate indicators. To date, GCs are used as standard biomarkers for distress [95,96]. However, due to the dual participation of GC in distress [97] and immune responses [98], and also because GC release can be triggered by a disease [99], it might not be feasible to clearly identify whether an animal is facing a distress or a disease challenge in the first place based solely on GC values.

One encounters a similar problem when trying to apply cytokines or acute phase proteins (APPs) as indicators for some infectious diseases or inflammation. Although there is accumulating evidence that APPs are valuable tools for the diagnosis of inflammation and infection, because they are a core part of the innate immune system, their release can also be triggered by distress [98]. Bürger et al. [100] suggested that in the absence of disease, APP might even serve as markers for distress, which supports the conclusion that they cannot be seen exclusively as signs for diseases. Both GC and APP undoubtedly have the potential to provide useful information about whether or not an issue exists. Nevertheless, because of their non-specificity and their intertwined relationship, based on their use alone, it might not be feasible to distinguish between disease and distress-related changes without a doubt. It is possible that specific release patterns or different concentrations of GCs and APPs related to a disease or distress response exist; however, to our knowledge, such explicit indications have not been discovered yet. In addition, the development of point-of-care diagnostics has not progressed enough to use molecular components as indicators. Nevertheless, wearable or implantable biosensors for the analysis of blood, saliva, or sweat [101] may have the potential to facilitate on-farm assessments of such components in the future.

However, distress and diseases also manifest on the non-molecular physiological and behavioural levels. On the clinical level, cytokine release causes changes in behaviour, which are collectively termed sickness behaviour [102]. Among lethargy and lack of appetite, increased and prolonged sleep is another characteristic of sickness behaviour [103]. Particularly, an increase in the slow waves during the non-rapid eye movement (NREM) sleep was observed in animals experiencing a health challenge [104]. By contrast, intense distress usually causes sleep disruption [105] and results in an overall reduction in sleep efficacy and the duration of slow-wave sleep [106]. Devices that monitor sleep-related changes, such as activity, heart rate or breathing [107], might facilitate the determination of the definite state an animal is experiencing. In addition, cytokines such as interleukin-1 elicit fever [103]. Although an increase in body temperature can also result from a distress response caused by heat or physical

activity, an increase over a prolonged period should be a good indicator for a disease-related change. By simultaneously monitoring environmental factors and e.g. activity, a reliable prediction could be made about the cause for the rise in body temperature. Finally, more specific clinical signs such as coughs are also good indicators for a disease-related challenge.

Another potential non-specific indicator could be grooming. Besides potentially allowing the characterization of ambivalent changes, as discussed in Recording of indicators, it might also facilitate the differentiation between distress- and disease-related changes. While grooming is markedly reduced during sickness [104], it can increase under distress due to its coping function [108,109]. These results encourage further studies on grooming, for example, in respect to the total amount of time spent grooming or to specific patterns related to e.g. the time of day or how the grooming is performed. Sensors such as brush monitoring systems [110,111] or cameras could enable the monitoring of grooming behaviour in animals. Finally, as previously discussed in the section 'Indicators for the assessment of welfare-related issues', changes in rhythmicity patterns of behavioural or physiological responses could also facilitate the differentiation between health and distress-related issues. Harper [112] showed that particularly the ultradian temperature rhythms at the third and fifth cycle per day exhibit a different pattern upon a social and a surgical challenge. Naturally, there might be countless other indicators for distress and disease, but as this would require a review of its own, it is beyond the scope of the current framework. Nevertheless, although the significance of new and presented indicators still needs to be investigated, it can be assumed that such indicators could complement the interpretation of molecular ones.

In summary, because the effects of disease and distress can affect or mask one another, differentiating between them is advantageous. Nonetheless, despite their intertwined relationship, distress is almost completely disregarded in studies on PLF technologies. Thus, there are hardly any defined indicators nor are there commercially available technologies, which allow the reliable identification of distress-related issues. Hence, we would like to encourage future research in this direction as we strongly believe that the differentiation between distress and disease is an important basis for the early detection of issues in livestock.

### (iii) Defined disease-related issues
#### Cause of disease

In the final and most specific decision level, the aim is to automatically detect signs related to defined diseases (figures 1 and 4). To do so, first, the determinants of the disease need to be identified. A very basic categorization can be made into infectious and non-infectious diseases. Injuries, intoxications, metabolic (e.g. ketosis) and genetic diseases are considered, among others, to be non-infectious diseases. By contrast, diseases elicited by pathogens such as bacteria, parasites or viruses are regarded as infectious diseases. For some defined diseases, such as mastitis, the identification of the causative agent is an important prerequisite for the instigation of the appropriate treatment [113]. Point-of-care diagnostics allowing the on-farm determination of the causative organism [62] will offer a great advantage in the diagnosis of defined diseases, though they are still under development.

#### Course of disease

Two major courses of disease exist, namely the acute and the chronic form [114]. An acute form is characterized by severe signs and a timely end of the disease and lasts approximately around 12–24 h [115] or 3–14 days [114]. The course of a chronic disease lasts around three months or more [115]. Differentiating between the two courses can be beneficial as they might require different medical treatments, which can vary, for example, in terms of frequency of drug administration or type of medication [116]. Acute and chronic diseases can exhibit clinical cases with clear recognizable signs [116] or subclinical cases generally without any clinically detectable signs [117]. A clinical disease has a considerable impact on productivity [118] and is related to an overall feeling of malaise [119,120]; therefore, the need for an early diagnosis is unquestionable. Although subclinical cases represent the light form of a disease, they can still negatively affect productivity [121] and the animal's welfare [122]. In addition, in the case of infectious diseases, the animals can still be contagious while lacking clear clinical signs. Further, over the course of time, a subclinical form can evolve into a clinical form [123]. All this makes the identification of subclinical diseases crucial. Of course, the detection of a subclinical disease is only possible in enterprises where an animal has a longer production life span and its health status is evaluated on an individual level.

#### Signs of disease

A variety of medical signs can be used for the evaluation of an issue related to a defined disease. Depending on which form of the disease is in focus, detection occurs at different levels. For clinical cases, the diagnostics are more straightforward because they can be determined by changes in the physical appearance of certain body regions (e.g. digestive system, eyes, ears, locomotor system) or tissues (e.g. mucosa). There is a range of PLF technologies commercially available, which are supposed to detect clinical signs. For example, some of the standard cardinal clinical signs for mastitis, such as redness and heat in the udder, can be recorded with thermal imaging systems (thermal camera, Agricam AB, Linköping, Sweden). In addition, sound analysis systems for cough detection (SoundTalks, SoundTalks NV, Leuven, Belgium) can nowadays be purchased. By contrast, subclinical cases can often be detected solely at the molecular level by taking and analysing samples of, for example, blood, milk or tissue, which is in most cases technically more challenging. For subclinical mastitis biochemical variables, cell count or bacteriological examinations serve as common indicators [124]. Also, APPs were proposed to reflect the presence of subclinical diseases [125]. While the development of biosensors such as for bacteriological or APP examination is still in progress, some milking systems for cows are able to analyse biochemical variables and determine the somatic cell count in the milk [3] and thus possibly also identify subclinical cases of mastitis or ketosis.

Regardless of which form of the disease should be detected, the focus in this decision level should primarily be on specific signs. The advantage of specific signs lies in their distinct nature, which might allow for the use of just one pathognomonic or a small number of meaningful signs

for diagnosis. Another advantage is that often a distinct threshold exists, which indicates a pathological state and makes calculating a reference value redundant. As an example, for subclinical ruminal acidosis, a pH value between 5.5 and 5.0 [126] and for acute ruminal acidosis a pH value < 5.0 [127] are seen as critical limits in the veterinary diagnosis. Further, the abundance of pathogens in the organism is a clear sign of disease. Despite the fact that only specific signs can clearly identify a distinct issue, they are usually accompanied by general systemic signs such as reduced feeding or activity. There are reported cases where specific signs appeared later than changes in general ones [29]. Further, changes in non-specific signs can even represent crucial alarm signs; for example, in ruminants, a stop in feed intake and rumination is dangerous and indicates a severe issue. However, despite these possible advantages and their function as supporting signs during a regular veterinary diagnosis, they are weak indicators for issues related to a defined disease. Additionally, in contrast to specific signs, a distinct threshold indicating pathological states rarely exists, hence calculating a reference value is here necessary to identify adverse states.

In summary, an important aspect to consider in this decision component is the type, course and form of disease. Further, the focus in this decision component should be placed on specific signs, which clearly indicate an issue related to a defined disease. Non-specific signs should only be taken into account, if at all, as supporting tools, which comply with the standard veterinary approach. Thus, on one hand, the last decision component is tied to the highest restrictions in terms of indicators. On the other hand, knowledge about the course or form of disease embodies the highest degree of information and might facilitate early initiation of the appropriate treatment.

## 5. Synthesis

Based on the requirements for each decision level presented in this framework, it becomes clear that most of today's available technologies, including those under development, are able to detect issues only related to (i) General welfare, which represents the first and broadest level of issue detection. Although distress is an important issue in livestock due to its effects on disease development and progression [91,92], and also due to its potential to mask signs of disease [93], there are hardly any systems which allow a clear differentiation between disease and distress-related changes. Finally, there are also just a few technologies available for the detection of signs related to a defined disease. In general, this shows that in many cases the expectations towards the technologies regarding outcome interpretation are actually too high as most systems only facilitate the detection of changes without knowing whether issues related to distress, general health or a defined disease are responsible for the changes. As a result, the term 'health alert' in PLF systems [3] is often misleading and raises expectations which might not be met, provided it is generated solely based on non-specific indicators, such as activity, feeding or weight.

Further, when looking at the potential indicators discussed in this framework that might be useful or necessary

for distinguishing between the three proposed decision levels, another question arises, namely whether the proposed indicators can be applied under current husbandry systems. Rhythmicity presents a good example for this point. A prerequisite for the function as an indicator is that disturbances such as management procedures or husbandry conditions do not considerably disrupt the rhythm of the animal, or even disrupt it to such an extent that a rhythm is no longer apparent or artificially imposed on the animals. Further, appropriate conditions such as a high space allowance or enrichment items might be required if luxury behaviours are going to be used. As a result, to reach the next level of PLF technology development, which would be characterized by enhanced and specified information, it might be necessary to rethink and adapt the concept of the present husbandry systems. Importantly, such a change might even reduce the occurrence and in turn the overlapping effects of issues and thereby make the distinction e.g. between disease- or distress-related issues more feasible

## 6. General conclusion

The presented framework emphasizes the need to specify the aim of detection and to choose appropriate indicators related to the purpose. Further, the framework highlights the key considerations, requirements and challenges regarding data collection for each decision level and can thus be used as a practical guide for the development of PLF technologies for the early detection of issues in livestock. Looking at the criteria proposed in the framework for the three decision levels and the detection targets set in commercially available technologies and studies, there seems to be a significant gap between the expectations for PLF technologies and the actual possibilities. Thus, it is of great importance that researchers and engineers carefully reflect on what conclusions can really be drawn from the collected data. In addition, it is also necessary to clarify the possibilities and restrictions of technologies during the commercialization process so that the end users are aware of how the outputs and alarms need to be interpreted. This is a crucial aspect if the technologies are applied on commercial farms or by policymakers to automatically examine the compliance of welfare regulations in livestock production. Finally, new indicators or indicator combinations that are on the horizon were discussed and should be explored to drive progress in the development of PLF technologies and enable more precise issue detection as proposed by the presented framework.

Data accessibility. This article has no additional data.

Authors' contributions. Both authors contributed equally to the manuscript. All authors gave final approval for publication and agree to be held accountable for the work performed therein.

Competing interests. We declare that we have no competing interests.

Funding. This work was supported by the Swiss Federal Food Safety and Veterinary Office and the Swiss Federal Office for Agriculture [1.18.14TG]. The content of the paper is solely the responsibility of the authors and does not necessarily represent the official views of the Swiss Federal Offices.

Acknowledgements. We thank Dr Sabine Gebhardt, Dr Beat Thomann, Dr Felix Adrion, Sibylle Zwygart, Dr Thomas Echtermann and Barbara Lutz for providing valuable comments on the manuscript.

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
