## [Peer Review File · Proceedings of the Royal Society B: Biological Sciences]

Review History

RSPB-2020-2288.R0 (Original submission)

Review form: Reviewer 1

Recommendation

Accept with minor revision (please list in comments)

Scientific importance: Is the manuscript an original and important contribution to its field?

Excellent

General interest: Is the paper of sufficient general interest?

Good

Quality of the paper: Is the overall quality of the paper suitable?

Good

Is the length of the paper justified?

Yes

Should the paper be seen by a specialist statistical reviewer?

No

Do you have any concerns about statistical analyses in this paper? If so, please specify them explicitly in your report.

No

It is a condition of publication that authors make their supporting data, code and materials available - either as supplementary material or hosted in an external repository. Please rate, if applicable, the supporting data on the following criteria.

Is it accessible?

N/A

Is it clear?

N/A

Is it adequate?

N/A

Do you have any ethical concerns with this paper?

No

Comments to the Author

Summary

This evidence synthesis article focusses on precision livestock farming (PLF) technologies for the early detection of welfare issues. The authors argue that, though there is a high demand in the livestock farming industry for commercialisation of PLF, so far only few tools with sufficient accuracy have been developed. They suggest that this is due to the tendency of specialised PLF to rely on indicators that are too general, implying that these indicators could be symptomatic for a wide range of issues and effectively being not reliable to detect highly specific problems. This results in these tools overpromising what they can deliver for commercial farming and also holding up the further development of more precise PLF technologies. The authors argue that first the intended purpose of a specialised PLF tool need to be carefully specified before selecting appropriate indicators tailored to the intended purpose. To address these issues the authors developed a decision-making framework with a focus on the data collection aspect that strives to improve the accuracy of PLF technologies. The main content of the manuscript delivers (i) the description of the developed three-stage decision-making framework to guide the automatic assessment of welfare problems at different levels of specificity (general welfare, diseases and distress, defined disease) for PLF technologies, and (ii) a rigorously researched review of PLF technologies for the early detection of welfare- and health-related issues relevant to the proposed framework. In my view, this is an important paper that not only gives a comprehensive, clearly documented overview on PLF technologies and their challenges for assessing animal health and welfare concerns but also offers a recommendation on how to improve their development by using the proposed framework, which will appeal to policy makers and stakeholders in livestock management alike.

Strength

The main strength of this paper is the proposed multidisciplinary decision framework, particularly because the authors involved experts from the respective disciplines to critically assess and verify each decision-stage. Also, the methods were clearly presented and the review itself rigorously integrated the results of previous studies in an unbiased way.

Critique

There are a few sections I would like the authors to revise or elaborate on, which are listed below. Unfortunately, line numbers were only available from page 19, but I try my best to make it clear

which paragraphs I'm referring to by posting the original sentence below each comment:

- Page 3, paragraph 2:

Comment (1): I believe the definitions of specificity and sensitivity may have been accidentally swapped. Specificity is the ability of a model / test to correctly identify the true negative rate, while sensitivity is the ability of a model / test to correctly identify the true positive rate.

"The upper values obtained for specificity (model's ability to detect positive outcome) found in the literature range around 86% for calving [24], 99.3% for oestrus [25], 88.84% for lameness [26] and 84.1% for mastitis [27]; for sensitivity (model's ability to detect negative outcomes), they range around 82% for calving [24], 91.3% for oestrus [25], 82.3% for mastitis [27] and for lameness 86.1% [23, 28]. However, high values of specificity and sensitivity can be achieved despite high error rates [23] and a considerable number of false-positive alerts [24]."

- Page 7, bottom of page:

Comment (2): When reading this for the first time I thought "cough detection based on coughs" sounds a bit confusing and doesn't explain what it actually means. I kindly suggest you change this to "...who tested an algorithm for automatic cough detection by classifying distinct cough sounds." Or something along these lines. Also, could you please elaborate what "any kind of problem" implies? Adding what it could refer to in brackets might be helpful (health, welfare, or productivity,...).

"For example, two of the few exceptions are the studies by Maselyne et al. [51], who used feeding-related variables for detecting any kind of problem, and by Guarino et al. [52], who tested an algorithm for cough detection based on coughs."

- Page 11

Comment (3): Something went wrong in the caption of Figure 2. I might have misunderstood, but aren't all free arrows pointing to empty spaces? Maybe you accidentally stated the same twice? I suspect that's the case. If not, I suppose I'm unsure what the free arrows are. Could you kindly clarify and modify the figure caption accordingly?

"Figure 2. Decision pathway for general welfare (I). The free arrows indicate that there are more options than presented. The arrows pointing to empty spaces indicate further possibilities."

- Page 20, line 49

Comment (4): A minor grammar mistake. Could you please add commas to "blood, saliva, or sweat".

- Page 21, line 68:

Comment (5): Regarding 'General welfare', I believe this should point to section I.

- Page 28, line 237 ff.:

Comment (6): This sentence is not completely clear to me, particular where you say "the term 'health alert' in PLF systems [20] is often misleading and raises expectations which might be met, provided...". It would make more sense to me if these expectation were "NOT met". Could you please elaborate or rephrase this sentence?

Review form: Reviewer 2

Recommendation

Reject – article is not of sufficient interest (we will consider a transfer to another journal)

Scientific importance: Is the manuscript an original and important contribution to its field?
Poor

General interest: Is the paper of sufficient general interest?
Marginal

Quality of the paper: Is the overall quality of the paper suitable?
Marginal

Is the length of the paper justified?
Yes

Should the paper be seen by a specialist statistical reviewer?
No

Do you have any concerns about statistical analyses in this paper? If so, please specify them explicitly in your report.
No

It is a condition of publication that authors make their supporting data, code and materials available - either as supplementary material or hosted in an external repository. Please rate, if applicable, the supporting data on the following criteria.

Is it accessible?
N/A

Is it clear?
N/A

Is it adequate?
N/A

Do you have any ethical concerns with this paper?
No

Comments to the Author

Review of RSPB-2020-2288

This manuscript aims to describe a framework for the development of PLF technologies for the early detection of welfare related problems in livestock by focusing on the decision process related to data collection. Using PLF is a fast increasing way to collect information of our livestock. Most PLF technologies are developed to manage production and are not developed to assess welfare at first. Nevertheless existing PLF technologies might collect useful information in relation to animal welfare, at the group and individual level. However at present many PLF technologies are hardly validated in a proper way which hampers application for reliable information collection.

Although the authors put a lot of emphasis in stressing all kind of relevant issues around PLF, these issues are well known. The novelty of their work should be the framework they propose. It does however in my opinion not give new insights in how to deal with information collected with PLF. An illustration that also the authors seem not really convinced by the value of the framework, is shown by the fact that the word framework is not coming back in the conclusion of the manuscript. In my view the study and framework does not provide any contribution to our knowledge on animal welfare in relation to PLF technology.

Some additional observations; currently animal welfare scientist are increasingly focusing on affective states as being the most relevant indicator for welfare: how do animals experience and feel about their own situation. Affective states but also positive indicators for welfare in general

are largely ignored in the given overview and framework. The authors seem to be most familiar with the veterinary side of welfare which results in an unbalanced view on animal welfare. The authors also make some awkward statements in relation to welfare which undermines the quality of the paper. Some examples: the authors state that 'behavioural problems or distress are also known to negatively affect the animals state and productivity'. Behavioural expressions are a result of the (mental and physical) state of the animal, not vice versa. The authors refer to the five freedoms in the Brambell report as being the concept of welfare. The five freedoms however were formulated by the FAWC many years after the publication of the report. In the chapter 'I General Welfare': 'adverse conditions resulting from compromised welfare' does turn around what is going on in reality; welfare can be compromised due to adverse conditions. The distinction between basic behaviour and luxury behaviour is artificial. Who determines what is basic and what is luxury for an animal. The authors' definition for basic behaviours is that they can be expressed under any circumstances regardless of the husbandry system or environmental or group effects. Given examples are feeding, rumination, and activity. But what about for example rooting behaviour in pigs? It might be seen as a luxury behaviour because pigs can find their food easily in a trough, but they perform rooting behaviour in any housing situation, so it can also be considered basic. This holds for almost any behaviour such as dustbathing, preening, grooming, play, social behaviour, etc. On the other hand stereotypies, an abnormal behaviour, can also be shown in many husbandry systems, but that should not be considered a basic behaviour. The conducted method of the literature review as described remains vague. It is a mixture of two methods, but how these methods are mixed is unclear. Studies that focus on development and validation of PLF were considered, but what kind of validation is meant? What were criteria to include a PLF technology or not? What quality check did they do on used grey literature, technical reports and proceedings? The last paragraph of the methods (a) section (line numbers are missing) is unclear. Three decision stages were identified; decisions on what? Experts were consulted, how many experts, what did they do to verify the framework, was anything changed or proposed to change by these experts? Many questions remain unanswered. Although the authors stress that animal based indicators should be used for animal welfare assessment, they propose in their framework to also include environmental indicators. Environmental indicator however do not contribute to the assessment of welfare. They might indicate the cause of a problem but not a problem itself.

Decision letter (RSPB-2020-2288.R0)

04-Nov-2020

Dear Dr Stachowicz:

I am writing to inform you that your manuscript RSPB-2020-2288 entitled "Do we automatically detect health or general welfare problems? A framework" has, in its current form, been rejected for publication in Proceedings B.

This action has been taken on the advice of referees, who have recommended that substantial revisions are necessary. With this in mind we would be happy to consider a resubmission, provided the comments of the referees are fully addressed. However please note that this is not a provisional acceptance.

Sincerely,
The Proceedings B Team
mailto: proceedingsb@royalsociety.org

Associate Editor
Board Member: 1
Comments to Author:

Thank you for submitting your interesting manuscript for consideration as a PRSB Evidence Synthesis article. Your work has now been reviewed by 2 experts in the field, and I have read the manuscript myself. While there is a consensus that the topic is pertinent for an evidence synthesis article, and timely in that it is likely to command broad interest, I am unable to take forward the current manuscript. You will see that a range of different views and recommendations have been proposed, not unusual in peer review, though collectively, your manuscript and topic, is in my opinion worthy of additional investment and consideration. As such, I encourage you strongly to consider the comments below, and submit a new manuscript at your earliest convenience. You will see that there are a variety of concerns which I echo, and highlight here, though full details are provided in the respective referee reports. For example, referee # 1 points out that there is a need to reconsider various definitions of key terms, as well as a need to review the utility of values (%) proposed for specificity. There is a lack of clarity in certain terms used, and further work is required in the caption of figure 2, and importantly, the decision pathway for the framework remains unclear. additional substantive issues are raised by referee# 2. Foremost among these is the claim that the proposed framework and your consideration based on the evidence presented, is not sufficiently novel. This of course is a potential flaw, and I would ask a specific and considered response both in terms of the uploaded response letter, but importantly an explicit consideration and statement of those elements that you find or propose is particularly novel is likely to advance the utility of such an approach. There is also concerns about the so-called affective states, and there needs to be an improved usage of terms used such as general welfare, and the outputs from the Brambell report. Of particular note again, is the expressed concern, that I echo, that the methodology for choice of literature and representation, is not sufficiently detailed or explicit. There is ambiguity in the choice of literature considered, with a claim that there is an over emphasis on the veterinary perspective. As you will know, it is crucial in evidence synthesis manuscripts, to ensure that not only is the literature base and evidence presented, fully accessible and transparent, but importantly, is robust, objective and transparent. Even if qualitative measures are proposed, it is important to understand how and why particular strategies for inclusion were implemented. There is also a lack of clarity in terms of the animal and environmental based indicators.

I would like to draw your attention specifically to our requirements for publication of Evidence Synthesis articles. Notwithstanding, in your response to referees, I would be grateful if you would include a brief account relating to my Editorial comments, on how the manuscript has been modified in relation to my brief suggestions. In particular, as you will have seen from the guidelines available for our Evidence Synthesis articles (<https://royalsocietypublishing.org/rspb/evidence-synthesis>), it is vital that the reader is able to assess the validity, robustness and objectivity of the evidence base presented. I would therefore appreciate a brief account of how the literature base presented has been selected (albeit in a qualitative fashion) as detailed in the link above), and to what degree you have been able to secure the appropriate level of representation, objectivity and standardisation in studies cited. Importantly also, when putting the final touches to the article, please ensure wherever possible, that where relevant, you have addressed some of the questions below, that characterises the Evidence Synthesis article type, though I fully recognise, that many questions will only partially apply to your manuscript :

1. Is the key policy-related question(s) articulated clearly?
2. Is there a clear justification in support of policy relevance?
3. Is the likely target audience identified clearly?
4. Does the search for literature utilise a comprehensive range of sources?
5. Does the synthesis article apply clearly documented inclusion criteria to all potentially relevant studies found during the search?
6. Is a clear methodology described to avoid bias?
7. Is your study objectively weighted according to methodological quality of cited literature?
8. Are knowledge gaps and priorities clearly identified?
9. Are outcomes/recommendations tangible in terms of likely impact?
10. Are all necessary supporting information available and accessible??

Including a brief indication of how you have addressed the specific criteria above in your response letter would be most helpful. I appreciate that the volume of revision is extensive, and may go beyond what you had originally anticipated. Notwithstanding, I would hope you will find the constructive and detailed suggestions helpful in formulating a more robust and representative evidence synthesis article for resubmission. As indicated below, as in all peer review processes, the invitation to resubmit, is of course no guarantee of eventual publication, but I will do my best to exercise consistency in the remaining peer review process, by approaching the original referees, at a minimum, though of course I am not in a position to confirm their availability.

Thank you in advance for bringing this information together, and we look forward to receiving the resubmitted manuscript in due course.

Reviewer(s)' Comments to Author:

Referee: 1

Comments to the Author(s)

Summary

This evidence synthesis article focusses on precision livestock farming (PLF) technologies for the early detection of welfare issues. The authors argue that, though there is a high demand in the livestock farming industry for commercialisation of PLF, so far only few tools with sufficient accuracy have been developed. They suggest that this is due to the tendency of specialised PLF to rely on indicators that are too general, implying that these indicators could be symptomatic for a wide range of issues and effectively being not reliable to detect highly specific problems. This results in these tools overpromising what they can deliver for commercial farming and also holding up the further development of more precise PLF technologies. The authors argue that first the intended purpose of a specialised PLF tool need to be carefully specified before selecting appropriate indicators tailored to the intended purpose. To address these issues the authors

developed a decision-making framework with a focus on the data collection aspect that strives to improve the accuracy of PLF technologies. The main content of the manuscript delivers (i) the description of the developed three-stage decision-making framework to guide the automatic assessment of welfare problems at different levels of specificity (general welfare, diseases and distress, defined disease) for PLF technologies, and (ii) a rigorously researched review of PLF technologies for the early detection of welfare- and health-related issues relevant to the proposed framework. In my view, this is an important paper that not only gives a comprehensive, clearly documented overview on PLF technologies and their challenges for assessing animal health and welfare concerns but also offers a recommendation on how to improve their development by using the proposed framework, which will appeal to policy makers and stakeholders in livestock management alike.

Strength

The main strength of this paper is the proposed multidisciplinary decision framework, particularly because the authors involved experts from the respective disciplines to critically assess and verify each decision-stage. Also, the methods were clearly presented and the review itself rigorously integrated the results of previous studies in an unbiased way.

Critique

There are a few sections I would like the authors to revise or elaborate on, which are listed below. Unfortunately, line numbers were only available from page 19, but I try my best to make it clear which paragraphs I'm referring to by posting the original sentence below each comment:

- Page 3, paragraph 2:

Comment (1): I believe the definitions of specificity and sensitivity may have been accidentally swapped. Specificity is the ability of a model / test to correctly identify the true negative rate, while sensitivity is the ability of a model / test to correctly identify the true positive rate.

"The upper values obtained for specificity (model's ability to detect positive outcome) found in the literature range around 86% for calving [24], 99.3% for oestrus [25], 88.84% for lameness [26] and 84.1% for mastitis [27]; for sensitivity (model's ability to detect negative outcomes), they range around 82% for calving [24], 91.3% for oestrus [25], 82.3% for mastitis [27] and for lameness 86.1% [23, 28]. However, high values of specificity and sensitivity can be achieved despite high error rates [23] and a considerable number of false-positive alerts [24]."

- Page 7, bottom of page:

Comment (2): When reading this for the first time I thought "cough detection based on coughs" sounds a bit confusing and doesn't explain what it actually means. I kindly suggest you change this to "...who tested an algorithm for automatic cough detection by classifying distinct cough sounds." Or something along these lines. Also, could you please elaborate what "any kind of problem" implies? Adding what it could refer to in brackets might be helpful (health, welfare, or productivity,...).

"For example, two of the few exceptions are the studies by Maselyne et al. [51], who used feeding-related variables for detecting any kind of problem, and by Guarino et al. [52], who tested an algorithm for cough detection based on coughs."

- Page 11

Comment (3): Something went wrong in the caption of Figure 2. I might have misunderstood, but aren't all free arrows pointing to empty spaces? Maybe you accidentally stated the same twice? I suspect that's the case. If not, I suppose I'm unsure what the free arrows are. Could you kindly clarify and modify the figure caption accordingly?

“Figure 2. Decision pathway for general welfare (I). The free arrows indicate that there are more options than presented. The arrows pointing to empty spaces indicate further possibilities.”

- Page 20, line 49

Comment (4): A minor grammar mistake. Could you please add commas to “blood, saliva, or sweat”.

- Page 21, line 68:

Comment (5): Regarding ‘General welfare’, I believe this should point to section I.

- Page 28, line 237 ff.:

Comment (6): This sentence is not completely clear to me, particular where you say “the term ‘health alert’ in PLF systems [20] is often misleading and raises expectations which might be met, provided...”. It would make more sense to me if these expectation were “NOT met”. Could you please elaborate or rephrase this sentence?

Referee: 2

Comments to the Author(s)

Review of RSPB-2020-2288

This manuscript aims to describe a framework for the development of PLF technologies for the early detection of welfare related problems in livestock by focusing on the decision process related to data collection. Using PLF is a fast increasing way to collect information of our livestock. Most PLF technologies are developed to manage production and are not developed to assess welfare at first. Nevertheless existing PLF technologies might collect useful information in relation to animal welfare, at the group and individual level. However at present many PLF technologies are hardly validated in a proper way which hampers application for reliable information collection.

Although the authors put a lot of emphasis in stressing all kind of relevant issues around PLF, these issues are well known. The novelty of their work should be the framework they propose. It does however in my opinion not give new insights in how to deal with information collected with PLF. An illustration that also the authors seem not really convinced by the value of the framework, is shown by the fact that the word framework is not coming back in the conclusion of the manuscript. In my view the study and framework does not provide any contribution to our knowledge on animal welfare in relation to PLF technology.

Some additional observations; currently animal welfare scientist are increasingly focusing on affective states as being the most relevant indicator for welfare: how do animals experience and feel about their own situation. Affective states but also positive indicators for welfare in general are largely ignored in the given overview and framework. The authors seem to be most familiar with the veterinary side of welfare which results in an unbalanced view on animal welfare. The authors also make some awkward statements in relation to welfare which undermines the quality of the paper. Some examples: the authors state that ‘behavioural problems or distress are also known to negatively affect the animals state and productivity’. Behavioural expressions are a result of the (mental and physical) state of the animal, not vice versa. The authors refer to the five freedoms in the Brambell report as being the concept of welfare. The five freedoms however were formulated by the FAWC many years after the publication of the report. In the chapter ‘I General Welfare’: ‘adverse conditions resulting from compromised welfare’ does turn around what is going on in reality; welfare can be compromised due to adverse conditions. The distinction between basic behaviour and luxury behaviour is artificial. Who determines what is basic and what is luxury for an animal. The authors’ definition for basic behaviours is that they can be expressed under any circumstances regardless of the husbandry system or environmental or group effects. Given examples are feeding, rumination, and activity. But what about for example rooting behaviour in pigs? It might be seen as a luxury behaviour because pigs can find their food easily in a trough, but they perform rooting behaviour in any housing situation, so it can also be considered basic. This holds for almost any behaviour such as dustbathing, preening, grooming,

play, social behaviour, etc. On the other hand stereotypical, an abnormal behaviour, can also be shown in many husbandry systems, but that should not be considered a basic behaviour. The conducted method of the literature review as described remains vague. It is a mixture of two methods, but how these methods are mixed is unclear. Studies that focus on development and validation of PLF were considered, but what kind of validation is meant? What were criteria to include a PLF technology or not? What quality check did they do on used grey literature, technical reports and proceedings? The last paragraph of the methods (a) section (line numbers are missing) is unclear. Three decision stages were identified; decisions on what? Experts were consulted, how many experts, what did they do to verify the framework, was anything changed or proposed to change by these experts? Many questions remain unanswered. Although the authors stress that animal based indicators should be used for animal welfare assessment, they propose in their framework to also include environmental indicators. Environmental indicators however do not contribute to the assessment of welfare. They might indicate the cause of a problem but not a problem itself.

Author's Response to Decision Letter for (RSPB-2020-2288.R0)

See Appendix A.

RSPB-2021-0190.R0

Review form: Reviewer 1

Recommendation

Accept as is

Scientific importance: Is the manuscript an original and important contribution to its field?

Good

General interest: Is the paper of sufficient general interest?

Good

Quality of the paper: Is the overall quality of the paper suitable?

Good

Is the length of the paper justified?

Yes

Should the paper be seen by a specialist statistical reviewer?

No

Do you have any concerns about statistical analyses in this paper? If so, please specify them explicitly in your report.

No

It is a condition of publication that authors make their supporting data, code and materials available - either as supplementary material or hosted in an external repository. Please rate, if applicable, the supporting data on the following criteria.

Is it accessible?

Yes

Is it clear?

Yes

Is it adequate?

Yes

Do you have any ethical concerns with this paper?

No

Comments to the Author

The paper has been much improved and I can endorse it for publication.

Decision letter (RSPB-2021-0190.R0)

13-Apr-2021

Dear Dr Stachowicz

I am pleased to inform you that your manuscript RSPB-2021-0190 entitled "Do we automatically detect health- or general welfare-related issues? A framework" has been accepted for publication in Proceedings B.

The referee(s) have recommended publication, but also suggest some minor revisions to your manuscript. Therefore, I invite you to respond to the referee(s)' comments and revise your manuscript. Because the schedule for publication is very tight, it is a condition of publication that you submit the revised version of your manuscript within 7 days. If you do not think you will be able to meet this date please let us know.

- 1) A text file of the manuscript (doc, txt, rtf or tex), including the references, tables (including captions) and figure captions. Please remove any tracked changes from the text before submission. PDF files are not an accepted format for the "Main Document".
- 2) A separate electronic file of each figure (tiff, EPS or print-quality PDF preferred). The format should be produced directly from original creation package, or original software format. PowerPoint files are not accepted.

3) Electronic supplementary material: this should be contained in a separate file and where possible, all ESM should be combined into a single file. All supplementary materials accompanying an accepted article will be treated as in their final form. They will be published alongside the paper on the journal website and posted on the online figshare repository. Files on figshare will be made available approximately one week before the accompanying article so that the supplementary material can be attributed a unique DOI.

Sincerely,

Professor Gary Carvalho

Comments to Author:

Thank you sincerely for the detailed response to the previous round of referee comments, and the revisions made in your resubmitted evidence synthesis article. As you will see, the referee who I contacted is happy with the changes made, and I have also gone through both your resubmitted manuscript, and your comprehensive and informative response letter. I am now fully satisfied with two of the initial crucial elements, concerning a prospective lack of novelty, and the methodology employed to select a sufficiently robust, representative and transparent evidence base.

The only suggestion I would make, though it is your decision to implement, as you see appropriate. I did find the additional information in your uploaded response letter, that is Figure 1, and Table 1, very informative, in terms of the strategy and choice of data for the systematic review. My own feeling is that it would be helpful to incorporate these in the supplementary information, clearly, with the appropriate cross-referencing in the main text. As I say, this is not essential, but I think would not only help to guide the readership, but would serve as a valuable template, for subsequent submissions of evidence synthesis articles. I do appreciate the time taken both in the revisions made, and in the constructive nature of your response letter. Therefore, we are happy now to proceed with acceptance, and look forward to seeing this interesting article published.

Reviewer(s)' Comments to Author:

Referee: 1

Comments to the Author(s).

The paper has been much improved and I can endorse it for publication.

Author's Response to Decision Letter for (RSPB-2021-0190.R0)

See Appendix B.

Decision letter (RSPB-2021-0190.R1)

19-Apr-2021

Dear Dr Stachowicz

I am pleased to inform you that your manuscript entitled "Do we automatically detect health- or general welfare-related issues? A framework" has been accepted for publication in Proceedings B.

Your article has been estimated as being 14 pages long. Our Production Office will be able to confirm the exact length at proof stage.

Data Accessibility section

Open Access

Paper charges

Sincerely,

Appendix A

Reviewer(s)' Comments to Author:

Referee: 1

Comments to the Author(s)

Summary

This evidence synthesis article focusses on precision livestock farming (PLF) technologies for the early detection of welfare issues. The authors argue that, though there is a high demand in the livestock farming industry for commercialisation of PLF, so far only few tools with sufficient accuracy have been developed. They suggest that this is due to the tendency of specialised PLF to rely on indicators that are too general, implying that these indicators could be symptomatic for a wide range of issues and effectively being not reliable to detect highly specific. This results in these tools overpromising what they can deliver for commercial farming and also holding up the further development of more precise PLF technologies. The authors argue that first the intended purpose of a specialised PLF tool need to be carefully specified before selecting appropriate indicators tailored to the intended purpose. To address these issues the authors developed a decision-making framework with a focus on the data collection aspect that strives to improve the accuracy of PLF technologies. The main content of the manuscript delivers (i) the description of the developed three-stage decision-making framework to guide the automatic assessment of welfare problems at different levels of specificity (general welfare, diseases and distress, defined disease) for PLF technologies, and (ii) a rigorously researched review of PLF technologies for the early detection of welfare- and health-related issues relevant to the proposed framework. In my view, this is an important paper that not only gives a comprehensive, clearly documented overview on PLF technologies and their challenges for assessing animal health and welfare concerns but also offers a recommendation on how to improve their development by using the proposed framework, which will appeal to policy makers and stakeholders in livestock management alike.

Strength

The main strength of this paper is the proposed multidisciplinary decision framework, particularly because the authors involved experts from the respective disciplines to critically assess and verify each decision-stage. Also, the methods were clearly presented and the review itself rigorously integrated the results of previous studies in an unbiased way.

Answ. 1. We are grateful for your time reviewing our manuscript, for the very positive assessment and for providing valuable feedback.

Critique

There are a few sections I would like the authors to revise or elaborate on, which are listed below. Unfortunately, line numbers were only available from page 19, but I try my best to make it clear which paragraphs I'm referring to by posting the original sentence below each comment:

Answ. 2. We are very sorry for the missing lines. We added line numbers to all pages, where they have been missing.

Page 3, paragraph 2:

Comment (1): I believe the definitions of specificity and sensitivity may have been accidentally swapped. Specificity is the ability of a model / test to correctly identify the true negative rate, while sensitivity is the ability of a model / test to correctly identify the true positive rate.

“The upper values obtained for specificity (model’s ability to detect positive outcome) found in the literature range around 86% for calving [24], 99.3% for oestrus [25], 88.84% for lameness [26] and 84.1% for mastitis [27]; for sensitivity (model’s ability to detect negative outcomes), they range around 82% for calving [24], 91.3% for oestrus [25], 82.3% for mastitis [27] and for lameness 86.1% [23, 28]. However, high values of specificity and sensitivity can be achieved despite high error rates [23] and a considerable number of false-positive alerts [24].”

Answ. 4. Thank you for pointing this out, we now assigned the definitions of sensitivity and specificity correctly (Lines 67-68 and 71-72).

Page 7, bottom of page:

Comment (2): When reading this for the first time I thought "cough detection based on coughs" sounds a bit confusing and doesn't explain what it actually means. I kindly suggest you change this to "...who tested an algorithm for automatic cough detection by classifying distinct cough sounds." Or something along these lines. Also, could you please elaborate what "any kind of problem" implies? Adding what it could refer to in brackets might be helpful (health, welfare, or productivity,...).

“For example, two of the few exceptions are the studies by Maselyne et al. [51], who used feeding-related variables for detecting any kind of problem, and by Guarino et al. [52], who tested an algorithm for cough detection based on coughs.”

Answ. 5. Thank you for the example, we agree and changed the first sentence as suggested (Lines 274-275). We also agree upon the suggestion of the second sentence, however, we have replaced the references by another and thus the sentence has been deleted. Now we are referring to a study where the difference between cough sounds of infected and non-infected pigs has been investigated (Lines 275-276). In addition, we added another reference to the text (Lines 276-277).

Page 11

Comment (3): Something went wrong in the caption of Figure 2. I might have misunderstood, but aren't all free arrows pointing to empty spaces? Maybe you accidentally stated the same twice? I suspect that's the case. If not, I suppose I'm unsure what the free arrows are. Could you kindly clarify and modify the figure caption accordingly?

“Figure 2. Decision pathway for general welfare (I). The free arrows indicate that there are more options than presented. The arrows pointing to empty spaces indicate further possibilities.”

Answ. 6. Yes, it was stated twice, thank you for the remark. We deleted the second sentence (Lines 391-392, 565, 677-678).

Page 20, line 49 Comment (4): A minor grammar mistake. Could you please add commas to “blood, saliva, or sweat”.

Answ. 7. We added the commas, as suggested (Line 614).

Page 21, line 68:

Comment (5): Regarding ‘General welfare’, I believe this should point to section I.

Answ. 8. We understand why it makes the impression that the sentence refers to Section I, however, in Section I.II we discuss the different aspects of indicators which can be recorded and grooming patterns are given as an example there. To make this more clear, we deleted part of the sentence in Line 633-634.

Page 28, line 237 ff.:Comment (6): This sentence is not completely clear to me, particular where you say “the term ‘health alert’ in PLF systems [20] is often misleading and raises expectations which might be met, provided...”. It would make more sense to me if these expectation were “NOT met”. Could you please elaborate or rephrase this sentence?

Answ. 9. That is correct, the word ‘NOT’ is missing, thank you once again. We added the word ‘NOT’ to the sentence (Line 808).

Finally, because of the length restrictions, we had to shorten our manuscript. Thus, text in the following lines was deleted: 92-95, 97, 103, 284-285, 291-292, 296-322, 324, 328, 335, 374-383, 398-399, 405-408, 410-411, 413, 417-422, 426, 434, 447-452, 462-468, 471-472, 475-477, 479-480, 493-495, 507-512, 520-522, 525-528, 532-534, 543-544, 547-548, 628-629, 636-637, 649, 669-671, 680-691, 750-757, 762-763, 767-793, 815-816, 819.

Referee: 2

Comments to the Author(s)Review of RSPB-2020-2288This manuscript aims to describe a framework for the development of PLF technologies for the early detection of welfare related problems in livestock by focusing on the decision process related to data collection.

Answ. 10. Thank you for reviewing our manuscript and for providing valuable comments.

Using PLF is a fast increasing way to collect information of our livestock. Most PLF technologies are developed to manage production and are not developed to assess welfare at first. Nevertheless, existing PLF technologies might collect useful information in relation to animal welfare, at the group and individual level. However at present many PLF technologies are hardly validated in a proper way which hampers application for reliable information collection. Although the authors put a lot of emphasis in stressing all kind of relevant issues around PLF, these issues are well known.

The novelty of their work should be the framework they propose. It does however in my opinion not give new insights in how to deal with information collected with PLF.

Answ.11. We disagree with the statement about the lack of novelty in our framework. While many previous studies have already discussed the insufficient performance of PLF technologies, they focused mostly on technical difficulties, the appropriate algorithms or measures for accuracy. In contrast, our framework is the first one to direct the attention to the very first step of system development, namely data collection. In our framework, we identified a major weakness of previous approaches, which is the arbitrary use of ambiguous indicators for a defined problem. A high number of studies claim to detect a specific problem while being only able to detect a potential problem related to any kind of stressor, e.g. general disease, a defined disease or psychological or thermal distress. Thus, in our framework we stress that it is not enough to only reflect on whether an indicator might be related to an issue in focus, but that it should be rather considered what other circumstances or issues might also lead to a change in the given indicator. To our knowledge, there has been no other paper, which has dealt with this issue and highlighted the ambiguity of the commonly used indicators. The need to discuss such issues is reflected by the many studies applying non-specific indicators for the detection of specific issues. In fact, based on our literature research from the 100 studies used for synthesis only 18 used a more specific approach.

To improve the predictive outcomes, it is of utmost importance to differentiate between issues and carefully reflect on whether the used indicators are closely related to the issue in focus. However, most studies ignore these aspects, which is not just mirrored in the use of non-specific indicators for a defined issue, but also in the neglect of the intertwined nature between distress and disease. Thus, in our framework we propose to detangle the aim of detection based on the degree of information into three decision levels I) general welfare-related issues, II) health- and distress-related issues and III) defined disease-related issues. The proposed decision levels help the reader to understand what level of detection can be achieved with the available data.

Further, for each of the proposed aims we provide a decision pathway, in which important considerations, challenges as well as the requirements for each decision levels are discussed. With this, we offer a multidisciplinary review about the areas animal behaviour and welfare, veterinary science and agricultural engineering and provide a practical guide for the development of PLF technologies. Further, we review potential new indicators such as rhythmicity or sleep, which might facilitate a more specified issue detection, but which to date remain largely overlooked. As a result, we strongly believe that our framework will positively contribute to the development and validation of PLF technologies for the automated assessment of issues in livestock.

An illustration that also the authors seem not really convinced by the value of the framework, is shown by the fact that the word framework is not coming back in the conclusion of the manuscript. In my view the study and framework does not provide any contribution to our knowledge on animal welfare in relation to PLF technology.

Answ. 12. We re-phrased the conclusion paragraph to make it clearer that the statements are derived from the framework (Lines 827-847). However, we would like to point out that we are convinced about the valuable contribution that the framework can bring to the table.

Some additional observations; currently animal welfare scientist are increasingly focusing on affective states as being the most relevant indicator for welfare: how do animals experience and feel about their own situation. Affective states but also positive indicators for welfare in general are largely ignored in the given overview and framework.

Answ. 13. The aim of our framework was not to discuss how a complete animal welfare assessment can be done. This has been discussed many times before and different concepts have evolved or have been further developed over the years. We agree that in the concept of Fraser (2007), but also in the more recent five freedoms, affective states are considered as an important component of animal welfare. Thus, affective states can indeed indicate welfare-related issues. However, to assess the affective state, actual indicators such as behavioural responses are needed. In our framework, the aim was to review these actual indicators, which can be automatically recorded for the detection of issues related to general welfare.

It is also important to point out that we did neither focus specifically on positive nor on negative indicators, because in many cases the same indicators can indicate a positive or a negative state. Instead, we provided a summary of existing indicators for the automated assessment of the animal's state.

It is unfortunate that we failed to communicate our purpose of the framework in a non-ambiguous way, which might be attributed to our phrasing, the lack of clear

explanations in some parts of the manuscript and some of the provided examples. Therefore, we made changes in the following lines 14-16, 27-32, 39-51, 61, 69-70, 72-73, 81-82, 219-244, 246-258, 271-272, 277-282, 323-324, 361-374, 384-385, 396-397, 483, 534-536, 553-554, 586-589 and we changed the headings of the three decision levels into general welfare-related issues, disease- and distress-related issues and issues related to defined diseases throughout the document. In addition, we changed the words unspecific to non-specific, problems to issues and stage (of detection) to level.

The authors seem to be most familiar with the veterinary side of welfare which results in an unbalanced view on animal welfare.

Answ. 14. Thank you for attributing a high familiarity with the veterinary science to us. As our expertise lies in the fields of animal behaviour and welfare and agricultural engineering we put a lot of effort into researching the area of veterinary science so that the manuscript demonstrates a balanced view on the three different disciplines.

The authors also make some awkward statements in relation to welfare which undermines the quality of the paper. Some examples: the authors state that 'behavioural issues or distress are also known to negatively affect the animals state and productivity'. Behavioural expressions are a result of the (mental and physical) state of the animal, not vice versa.

Answ. 15. Thank you for the remark. We agree that the sentence was not properly phrased and deleted it (Lines 39-40) and re-phrased part of the respective paragraph.

The authors refer to the five freedoms in the Brambell report as being the concept of welfare. The five freedoms however were formulated by the FAWC many years after the publication of the report.

Answ. 16. The first postulated freedoms "freedom to stand up, lie down, turn around, groom themselves and stretch their limbs" were formulated by the Brambell committee and based on the Brambell report the FAWC re-elaborated the concept of the 5 freedoms. We agree, however, that the phrasing and the lack of explanation is a source of confusion. To avoid further confusion about the aim of our framework in relation to welfare assessment (see Answer 13), we deleted the respective sentence in the introduction (Lines 43-44).

In the chapter 'I General Welfare': 'adverse conditions resulting from compromised welfare' does turn around what is going on in reality; welfare can be compromised due to adverse conditions.

Answ. 17. Thank you for pointing this out. Again, we agree that there is a mistake in the sentence structure. We deleted the sentence (Lines 374-375) and re-phrased the previous sentences and paragraph (Lines 362-374).

The distinction between basic behaviour and luxury behaviour is artificial. Who determines what is basic and what is luxury for an animal. The authors' definition for basic behaviours is that they can be expressed under any circumstances regardless of the husbandry system or environmental or group effects. Given examples are feeding, rumination, and activity. But what about for example rooting behaviour in pigs? It might be seen as a luxury behaviour because pigs can find their food easily in a trough, but they perform rooting behaviour in any housing situation, so it can also be considered basic. This holds for almost any behaviour such as

dustbathing, preening, grooming, play, social behaviour, etc. On the other hand stereotypies, an abnormal behaviour, can also be shown in many husbandry systems, but that should not be considered a basic behaviour.

Answ. 18. We are sorry for the phrasing and the resulting confusion about the definition of basic (core) and luxury (low resilience) behaviours. The sentence in our manuscript mentioned by the reviewer does not represent the definition for luxury/low resilience or basic/core behaviours. With the sentence, we aimed for explaining that basic/core behaviours are usually easier to perform in husbandry systems, whereas for the expression of luxury/low resilience behaviours such as play or exploration usually more resources such as ample space or enrichment items are needed.

There are several papers, which have discussed or used the term basic/core and luxury/low resilience behaviours, for references see the following literature:

-Müller, Münger, Mandel, Eggerschwiler, Schwinn, Gross, Bruckmaier, Hess, Dohme-Meier (2018). Physiological and behavioural responses of grazing dairy cows to an acute metabolic challenge. *Journal of Animal Physiology and Animal Nutrition*, 1-11.

- Mandel, Whay, Nicol, Klement (2013). The effect of food location, heat load, and intrusive medical procedures on brushing activity in dairy cows. *Journal of dairy science* 96, 6506-6513.

-Held and Spinka (2011). Animal play and animal welfare. *Animal Behaviour* 81, 891-899.

-Duncan (1998). Behavior and Behavioral Needs. *Poultry Science* 77, 1766-1772.

-Weary, Huzzey, von Keyserlingk (2009). BOARD-INVITED REVIEW: Using behavior to predict and identify ill health in animals. *Journal of Animal Science* 87, 770-777.

In general, basic/core behaviours are assumed to serve an immediate function and thus provide a short-term value. Luxury or low resilience behaviours on the other hand are expected to decrease when time or energy resources are limited, as they are thought to provide long-term benefits. Because of that, it is assumed that luxury behaviours are the better choice for assessing welfare-related issues in animals, as these will be the first ones to change when an animal is facing a challenge.

We re-phrased the paragraph and the definition of luxury behaviours and added the definition for core/basic behaviours (Lines 438-441, 443-446, 454-461).

The conducted method of the literature review as described remains vague. It is a mixture of two methods, but how these methods are mixed is unclear.

Answ. 19. We understand the difficulty with the mixed methods. We now decided to exclude the results from google scholar, as they provided only a few additional articles of grey literature. The advantage is that without the search in google scholar, we meet the definition of a systematic literature search. We re-phrased and added information to the methods section (Lines 108-175). In addition, we now provide a results section of the systematic literature research (Lines 200-217) and a list of the included studies for synthesis and the search protocol for the databases as supplemental material.

Studies that focus on development and validation of PLF were considered, but what kind of validation is meant?

Answ. 20. Studies were included, which validated or developed a technology or an algorithm for the detection of issues. For example, studies which used the recorded variables such as walking, lying and standing for the assessment of lameness. Excluded were studies, which validated or developed a technology for the basic

functioning of the system. For example, studies which validated whether standing, lying and walking can be classified based on an accelerometer. We re-phrased the sentence about the validation criteria (Lines 143-149 and 156-160).

What were criteria to include a PLF technology or not?

Answ. 21. We included all studies about PLF technologies which 1. were developed or validated for the purpose of the assessment of issues in livestock, see answer to previous comment. 2. PLF systems which have been developed for the following enterprises: dairy cows, beef cattle, calves, broilers, laying hens, goats and sheep, pigs and sows. 3. Studies written in English and 4. studies with negative and positive results and 5. Studies published at any year. We re-phrased the paragraph about the inclusion and exclusion criteria (Lines 142-161, see also previous answer).

What quality check did they do on used grey literature, technical reports and proceedings?

Answ. 22. We have excluded the search from google scholar. No grey literature citations remain in the manuscript. We do not class the proceedings from the ECPLF conferences as grey literature as they were peer reviewed and published with an ISBN number.

The last paragraph of the methods (a) section (line numbers are missing) is unclear. Three decision stages were identified; decisions on what?

Answ. 23. The mentioned paragraph is about the decision levels for problem detection and about the aspects and challenges discussed for each proposed decision level. For clarification, we re-phrased the paragraph (Lines 178-198).

Experts were consulted, how many experts, what did they do to verify the framework, was anything changed or proposed to change by these experts? Many questions remain unanswered.

Answ. 24. The focus group consisted of six experts (authors excluded). First, each of the experts reviewed the manuscript on its own, and then the comments and suggestions were discussed with all experts in a meeting. Based on the discussion, it has been decided on three major adjustments in the manuscript. 1. To provide a wider overview about the different challenges one faces when trying to develop PLF technologies in the introduction, including technical difficulties and methods for data analysis. 2. To improve the terminology of veterinary terms in the defined diseases section and 3. To add a discussion to the synthesis about the current expectation towards PLF technologies and what actually can be accomplished to date, under consideration of the criteria proposed in the framework. We added this information to the methods (Lines 190-198).

Although the authors stress that animal based indicators should be used for animal welfare assessment, they propose in their framework to also include environmental indicators. Environmental indicator however do not contribute to the assessment of welfare. They might indicate the cause of a problem but not a problem itself.

Answ.25: It is often stated that there are two kinds of indicators, which can serve for welfare assessment namely environment (resource)- and animal-based indicators, for references see the following literature:

-Leliveld and Provolo (2020). A Review of Welfare Indicators of Indoor-Housed

Dairy Cow as a Basis for Integrated Automatic Welfare Assessment Systems. *Animal*, 10, 1430.

-G. Stilwel (2016). Small ruminants' welfare assessment—Dairy goat as an example. *Small Ruminant Research* Volume 142, 51-54.

-Anonymous (2001). Scientists' Assessment of the Impact of Housing and Management on Animal Welfare, *Journal of Applied Animal Welfare Science*, 4, 3-52, DOI: 10.1207/S15327604JAWS0401_2

P. F. Johnsen , T. Johannesson & P. Sandøe (2001). Assessment of Farm Animal Welfare at Herd Level: Many Goals, Many Methods, *Acta Agriculturae Scandinavica, Section A - Animal Science*, 51, 26-33

We agree that animal-based indicators are better measures, because as already stated in the manuscript, they present direct responses of the animal. However, environmental indicators can provide additional information about the presence of potential stressors and, thus, help to identify the cause for a problem, which is a crucial aspect for the initiation of appropriate interventions in practice (Main et al., 2003). For example, given that a rise in body temperature is registered in an animal, by simultaneously measuring the environmental temperature, it might be possible to conclude whether an emerging infection or heat stress might be responsible for the change. Thus, we think it is very reasonable to consider environmental indicators as supporting tools for the assessment of welfare-related issues in livestock.

However, we re-phrased the sentence and added information to Lines 402-404 and 408-409 to make it clearer, that environmental indicators should be only used as supporting tools and that they have to be seen as a risk indicators for issues related to welfare and not as a sign for an existing problem related to welfare.

Finally, because of the length restrictions, we had to shorten our manuscript. Thus, text in the following lines was deleted: 92-95, 97, 103, 284-285, 291-292, 296-322, 324, 328, 335, 374-383, 398-399, 405-408, 410-411, 413, 417-422, 426, 434, 447-452, 462-468, 471-472, 475-477, 479-480, 493-495, 507-512, 520-522, 525-528, 532-534, 543-544, 547-548, 628-629, 636-637, 649, 669-671, 680-691, 750-757, 762-763, 767-793, 815-816, 819.

Appendix B

Revision Manuscript ID RSPB-2021-0190.R1

Comment: The only suggestion I would make, though it is your decision to implement, as you see appropriate. I did find the additional information in your uploaded response letter, that is Figure 1, and Table 1, very informative, in terms of the strategy and choice of data for the systematic review. My own feeling is that it would be helpful to incorporate these in the supplementary information, clearly, with the appropriate cross-referencing in the main text. As I say, this is not essential, but I think would not only help to guide the readership, but would serve as a valuable template, for subsequent submissions of evidence synthesis articles. I do appreciate the time taken both in the revisions made, and in the constructive nature of your response letter. Therefore, we are happy now to proceed with acceptance, and look forward to seeing this interesting article published.

Answer: We have added the adapted PRISMA flow diagram (Figure 1 in the response letter) to the supplementary material (as Figure 1SP) and inserted the cross-reference in the manuscript (Lines 154-155). However, Table 1 in the response letter is actually Table 1 in the manuscript (Line 104), thus there is no need to include the table also in the supplementary material.

In addition, we have updated Figure 2 and 3, as we have noticed some spelling mistakes (Lines 281 and 396).

Further, we wanted to include additional information to the sentence in Lines 590-594 and therefore we had to re-phrase it. The red part was deleted and the green part was added:

Old sentence: **In order to function as an indicator a prerequisite is that disturbances such as management procedures or housing conditions do not disturb the animal's rhythm considerably, or even to such an extent that a rhythm is no longer evident.**

Sentence re-phrased: **A prerequisite for the function as an indicator is that disturbances such as management measures or husbandry conditions do not considerably disrupt the rhythm of the animal or even disrupt it to such an extent that a rhythm is no longer apparent or is artificially imposed on the animals.**

Finally, during the re-submission process it was indicated that our abstract is too long and we had to shorten it by 29 words. Therefore, we deleted the sentence in Lines 26 to 28 and the word 'more' in Line 28. We also re-phrased the sentence in Lines 17-18.